# Phytochemicals as Regulators of Genes Involved in Synucleinopathies

**DOI:** 10.3390/biom11050624

**Published:** 2021-04-22

**Authors:** Andrei Surguchov, Libby Bernal, Alexei A. Surguchev

**Affiliations:** 1Department of Neurology, University of Kansas Medical Center, Kansas City, KS 66160, USA; 26808 Fonticello Street, Prairie Village, KS 66208, USA; libby.bernal@icloud.com; 3Section of Otolaryngology, Department of Surgery, Yale School of Medicine, Yale University, New Haven, CT 06520, USA; alexei.surguchev@yale.edu

**Keywords:** synucleinopathies, Parkinson’s disease, α-synuclein, epigenetics, neurodegeneration, protein aggregation, dementia with Lewy bodies, multiple system atrophy, DNA methylation, histone modifications, phytochemicals, dietary supplements

## Abstract

Synucleinopathies are a group of neurodegenerative diseases characterized by the accumulation of α-synuclein aggregates in neurons, nerve fibers or glial cells. Three main types of diseases belong to the synucleinopathies: Parkinson’s disease, dementia with Lewy bodies, and multiple system atrophy. All of them develop as a result of an interplay of genetic and environmental factors. Emerging evidence suggests that epigenetic mechanisms play an essential role in the development of synucleinopathies. Since there is no disease-modifying treatment for these disorders at this time, interest is growing in plant-derived chemicals as a potential treatment option. Phytochemicals are substances of plant origin that possess biological activity, which might have effects on human health. Phytochemicals with neuroprotective activity target different elements in pathogenic pathways due to their antioxidants, anti-inflammatory, and antiapoptotic properties, and ability to reduce cellular stress. Multiple recent studies demonstrate that the beneficial effects of phytochemicals may be explained by their ability to modulate the expression of genes implicated in synucleinopathies and other diseases. These substances may regulate transcription directly via transcription factors (TFs) or play the role of epigenetic regulators through their effect on histone modification, DNA methylation, and RNA-based mechanisms. Here, we summarize new data about the impact of phytochemicals on the pathogenesis of synucleinopathies through regulation of gene expression.

## 1. Introduction: Pathophysiology of Synucleinopathies

Synucleinopathies are a group of neurodegenerative diseases that pose a major health issue in an aging population. Since the proportion of older adults now is higher than ever before, the prevalence of these disorders is steadily growing. Synucleinopathies are characterized by the accumulation of α-synuclein (Figure 1A) aggregates in both neuronal and non-neuronal cells, primarily in the brain [1]. The group consists of three main types of illnesses: Parkinson’s disease (PD), dementia with Lewy bodies (DLB), and multiple system atrophy (MSA).

PD and DLB’s common feature is an abnormal accumulation of aggregated misfolded α-synuclein in proteinaceous neuronal inclusions named Lewy bodies (Figure 1B, C) and Lewy neurites [2,3,4,5]. In MSA, α-synuclein inclusions accumulated mostly within oligodendrocytes are called glial cytoplasmic inclusions (GCI) (Figure 1D) [6].

Genetic mutations, mitochondrial dysfunction, defects in proteolytic systems, and neuroinflammation contribute to their pathophysiology [4]. Although the three types of synucleinopathies share some common mechanisms, a growing body of evidence indicates the existence of structural and functional differences underlying these diseases’ pathogenesis. One of them is the presence of distinct α-synuclein strains in samples from patients with PD, DLB, and MSA [7,8]. There is no disease-modifying treatment for these disorders, restricting cure options to symptomatic relief measures and palliative procedures. Thus, developing new effective therapeutic strategies to control and cure these devastating diseases is an urgent need [9].

## 2. Parkinson’s Disease (PD)

PD is the second most prevalent neurodegenerative disease after Alzheimer’s disease. It is a progressive and severely debilitating disorder associated with decreasing dopamine-secreting neurons and Lewy bodies’ formation in the substantia nigra and basal ganglia [10]. PD is characterized by progressive tremor, rigidity, bradykinesia, and postural instability. More than 6 million people had PD in 2015 [11], and, according to estimates, this number will reach 20 million by the year 2050 [12,13]. Several mechanisms contribute to the PD pathogenesis, such as protein misfolding and mishandling, the impairment of mitochondrial functions, neuroinflammation, oxidative damage to neurons and autoimmunity [14,15,16]. Despite the availability of symptomatic treatments, methods for health improvement in PD patients remain scarce. The majority of the treatment methods ease the symptoms, including the relief of motor impairment, without addressing the disease’s basic causes, so there is no proven disease-modifying therapy.

### 2.1. Familial Cases of PD

Familial PD cases account for less than 10% of the overall incidence, and the remaining 90% are sporadic forms of the disease [17]. Several rare familial forms of PD have been described. The first one was identified in an Italian kindred and three unrelated families of Greek origin due to a mutation in the α-synuclein gene [18]. Other causative mutations in several other genes, for example, PARK2 (E3 ubiquitin protein ligase), PINK1 (PTEN-induced kinase 1), PARK7 (DJ1, deglycase), LRRK2 (leucine-rich repeat kinase), VPS35 (Vacuolar protein sorting ortholog 35, CHCHD2 (Coiled-Coil-Helix-Coiled-Coil-Helix Domain Containing 2) have been linked to familial forms of PD [16].

Approximately 19 PD-causing genes, including ten autosomal dominant and nine autosomal recessive ones, have been identified [19]. Notably, most genome-wide association study (GWAS) hits are driven by noncoding variations and most probably are associated with gene expression regulation [16,20]. Multiple lines of evidence suggest that the alterations in gene expression are associated with PD pathology, and epigenetic mechanisms play an essential role in the pathophysiology of this disorder [16].

### 2.2. Heritable Risk of PD

Classic Mendelian genetics cannot explain heritable risk, suggesting that other factors are involved [21,22]. Accumulating data indicate that epigenetics can shed light on unknown aspects of complex diseases like PD. Nutrition, including the consumption of phytochemicals, is an essential environmental factor related to PD through epigenetic pathways [16].

### 2.3. Environmental Factors in PD Pathogenesis

Some environmental factors reduce the risk of PD, whereas others may raise it. Many natural products, including the phytochemicals present in dietary products, decrease PD development risk (see Section 6). Nutrition might grow to become a preventive and even therapeutic alternative against PD, especially if combined with other interventions like L-3,4-dihydroxyphenylalanine (L-DOPA) and its derivatives. However, according to epidemiologic and toxicological data, there is a consistent association between exposure to herbicides, pesticides, and other exogenous toxicants (for example, paraquat or rotenone) and PD [23]. Lifestyle factors, head injury, and exposure to industrial toxicants, such as various solvents, or metals, also increase the risk of PD.

## 3. Dementia with Lewy Bodies (DLB)

DLB is an age-associated neurodegenerative disease characterized by a progressive cognitive decline that restricts daily activities and affects everyday life. DLB develops due to a combination of genetic variations and epigenetic changes, including DNA methylation and histone alterations modulating the level of gene expression [24]. DLB is associated with the accumulation of misfolded α-synuclein in Lewy bodies and Lewy neurites [25] (Figure 1C). This neuropathological feature of DLB is similar to PD. The GWAS indicated a number of associations with apoE, α-synuclein, glucocerebrosidase, and contactin 1 [25,26]. Cholinesterase inhibitors are most often used for DLB treatments, but they are not disease-modifying drugs, and they are unable to eliminate the main course of the disease. The absence of an effective DLB therapy may be explained by a lack of comprehensive understanding of the mechanism underlying the disease. There are no medications specifically for the treatment of DLB approved by the US Food and Drug Administration (FDA). Therefore, drugs developed and tested for AD and PD treatment are sometimes used for DLB therapy [27,28]. Thus, new drug targets should be identified to develop DLB-specific disease-modifying therapies.

## 4. Multiple System Atrophy (MSA)

MSA is a debilitating and fatal neurodegenerative disorder clinically characterized by a combination of parkinsonism, autonomic failure and cerebellar ataxia [29,30]. MSA may be considered a form of atypical parkinsonism that affects a patient’s motor ability and autonomic function. Pathologically MSA is characterized by neuronal loss and gliosis in the cerebellum or basal ganglia, which usually correlate with the clinical phenotype [31]. The causes of MSA are unclear, and therapies significantly improving the disease course have not been developed. Recent studies revealed several signs linking MSA pathogenesis with PD, including inflammation, α-synuclein aggregation in oligodendroglia, protein misfolding, impairment of the tubulin polymerization-promoting protein, and mitochondrial dysfunction [31,32,33,34]. A unique pathological hallmark of MSA is an accumulation of GCI containing α-synuclein in oligodendrocytes (Figure 1C). The GCI burden is a crucial factor in MSA pathology. Preventive strategies inhibiting GCI formation may overturn disease progression. Clinical trials of riluzole, growth hormone, minocycline, rifampicin, and epigallocatechin gallate have been conducted, but did not show noticeable benefits [35].

## 5. Role of Epigenetics in Synucleinopathies

Epigenetic modifications alter the gene function without changing the underlying DNA sequence [36,37].

Negatively charged DNA is wrapped around positively charged histones forming a nucleosome, a simple chromatin unit (Figure 2). Chemical alterations to histone proteins include methylation and acetylation. These histone modifications can induce the formation of an open DNA state. The open state enables gene expression by allowing TFs and enzymes to bind to DNA. Alternatively, chromatin may acquire a closed heterochromatin state suppressing gene expression by inhibiting the initiation of transcript [38]. These epigenetic modifications can be induced by various factors, including phytochemicals present in the diet. Epigenetic changes in gene function are heritable and are not attributed to alterations of the DNA sequence [39].

### 5.1. Epigenetic Mechanism and Neurodegeneration

DNA methylation and post-translational modifications of nucleosomal histones (Figure 2) alter chromatin structure and regulate gene expression. The study of epigenetic regulation has many exciting translational aspects, one of which is analysis of association explaining the mechanisms linking gene and diet. Investigation in this field can elucidate the relationship between nutrition and neurodegenerative diseases. Changes in the epigenome may cause transcriptional alterations and genomic instability, thus contributing to the development of neurodegenerative diseases and other age-related disorders [39]. Growing evidence supports the hypothesis that various epigenetic mechanisms such as DNA methylation, histone modifications, and noncoding RNAs (ncRNAs) are organized and coordinated, forming an “epigenetic network” [40].

### 5.2. Epigenetic Mechanisms in PD Pathogenesis

DNA methylation, histone modifications, and RNA-based mechanisms are the principal epigenetic mechanisms involved in PD pathophysiology (Figure 2). These biochemical processes regulate the expression of α–synuclein and other proteins involved in PD pathogenesis. One of the indirect proofs of the role of epigenetics in PD pathogenesis is the finding that DNA in PD patients has increased cytosine modifications, especially in genes responsible for neurogenesis, synaptic structure and neurodevelopment [21].

#### 5.2.1. DNA Methylation

DNA methylation is one of the best-characterized chromatin modifications. The human genome is mostly methylated on CG bases cytosine–phosphate–guanine motifs, which are enriched in promoters. The methylation of these regions by DNA methyltransferases (DNMTs) causes gene silencing. Abnormal DNA methylation is associated with the pathogenesis of PD. DNMTs are the enzymes for the establishment and maintenance of DNA methylation patterns. However, there is a hypothesis that DNA methylation may not be the most attractive target for the development of PD treatment [41]. Other mechanisms, such as global hyperacetylation, and histone deacetylase (HDAC) dependent regulation of α-synuclein expression may be more prominent targets from the current perspective. Corresponding histone modifiers may become convenient targets for small molecule inhibition therapeutics in the future [41].

#### 5.2.2. DNA Methylation Changes in PD

Several alterations in DNA methylation associated with PD pathogenesis are described. For example, DNMT1 catalyzing DNA methylation is downregulated in the postmortem brain samples of PD patients [42]. Although disputable due to the insufficiency of direct evidence [43], those changes may be associated with alterations in the gene expression of α-synuclein and other genes implicated in PD development, ultimately causing abnormal protein aggregation [42].

An important association of DNMT1 with PD pathogenesis was described by Desplats and coauthors [44]. They reported that DNMT1 is downregulated in the postmortem brain of PD patients [44]. A global 30% reduction in DNA methylation correlates with an increased level of α-synuclein and associated sequestering of DNMT1 outside the nucleus [44]. However, the study of the role of DNMT1 in the DNA methylation of the α-synuclein gene brought contradictory and inconclusive results concerning this process’s details [45]. It was reported that the protein expression of DNMT1 was reduced, whereas the corresponding mRNA levels increased in the cellular and mouse models of PD [46]. Further investigation identified a role of an additional regulator of the methylation process. As shown recently, miR-17 mediates DNMT1 downregulation and may be responsible for the aberrant DNA methylation in PD [46].

#### 5.2.3. Histone Acetylation

The imbalance of histone acetylation also plays an important role in PD pathogenesis. Several types of histone acetyltransferases and histone deacetylases are involved in regulating the acetylation of histone lysine residues [47]. For example, modification of the DNA packaging protein histone H3 (H3K27ac) dysregulates α-synuclein expression, one of the important players in PD.

There is an opinion that DNA hydroxymethylation, global hyperacetylation, and HDAC dependent regulation of α-synuclein expression play a more significant role in PD development than mechanisms directly controlling DNA methylation [41]. The treatment with DNMT inhibitors in vitro indicates that DNA methylation is a potent mechanism for α-synuclein downregulation. The best-studied regulator of α-synuclein expression is a CpG island in intron 1. Its efficiency is especially high in cells with a low endogenous level of α-synuclein expression [41].

#### 5.2.4. Other Epigenetic Mechanisms Associated with PD

Other important epigenetic players are the noncoding RNAs (ncRNAs). Recent genome-wide studies have shown that <2% of the human genome codes for proteins. However, the genome is extensively transcribed, producing regulatory noncoding RNAs (ncRNAs). Noncoding RNAs may be categorized in two main groups on the basis of their length. The most thoroughly studied microRNAs (miRNAs) are 20–22 nucleotide-long molecules that downregulate the gene expression on the post-transcriptional level, binding to mRNAs. Long noncoding RNAs (lncRNAs) have lengths of more than 200 nucleotides. They regulate gene expression through various mechanisms, most often at the epigenetic level [48]. LncRNAs exist in either a linear or a circular form (circRNAs). There is growing evidence that noncoding RNAs play a significant, but not yet completely understood role in the PD pathogenesis [49].

### 5.3. Role of Epigenetics in DLB

Funahashi et al. examined an association of CpG island methylation level in the α-synuclein gene with DLB [50]. The authors found that intron 1 methylation in leucocytes is significantly reduced in DLB patients compared to controls. Similar results indicating a reduction in DNA methylation in DLB patients were described by Desplats (2011) [44]. These authors found that hypomethylation is associated with the translocation of DNMT1 from the nucleus to the cytoplasm, similar to the findings described in PD patients [44]. Thus, similar epigenetic dysregulation occurs in DLB and PD.

In addition to the hypomethylation of the α-synuclein gene, the epigenetic regulation of other genes associated with DLB has been proposed, including ataxin 2 (ATXN2), deglycase PARK7 and parkin (PRKN) [24]. Furthermore, changes in H3 histone acetylation might also contribute to LBD development [24]. Since epigenetic modifications in DLB patients are reversible and dynamic, they are used as appropriate targets for DLB treatment.

### 5.4. Epigenetic Mechanisms in MSA

Several epigenetic mechanisms are associated with MSA pathogenesis. Rydbirk and coauthors have investigated 5-methylcytosine (5mC) and 5-hydroxymethylcytosine changes throughout the genome of MSA patients and control individuals [51]. They found five significantly different 5mC probes in the genome of MSA patients. One of these probes mapped to the apoptosis resistant E3 ubiquitin protein ligase 1 (AREL1) gene implicated in antigen presentation was reduced in MSA patients. This decrease correlated with elevated 5 mC levels. Several functional DNA methylation modules are involved in inflammatory processes. In the brains of MSA patients, reduced 5 mC levels on AREL1 were concordant with elevated level of gene expression of both AREL1 and MHC Class I HLA genes.

In another study consistent alterations in myelin-associated oligodendrocyte basic protein (MAOBP) and huntingtin interacting protein 1 (HIP1) DNA methylation status were described in MSA patients [52,53]. In particular, reduced MOBP mRNA levels correlated with elevated DNA methylation in MSA patients. A distinct association between the methylation of HIP1 DNA and gene expression levels has been found in MSA patients compared to healthy controls [52]. These results propose that this locus is subjected to epigenetic remodeling in MSA.

In another study, Bettencourt and coauthors [53] described DNA methylation changes across brain regions in MSA with different degrees of GCI pathology. We identified 157 CpG sites and 79 genomic regions where DNA methylation was significantly altered in the MSA cases. Some DNA methylation changes mirror the MSA-associated pathology, for example, cerebellum-specific alterations [53].

Thus, multiple lines of evidence indicate that epigenetic mechanisms play an important role in all three main forms of synucleinopathies: PD, DLB and MSA. Accumulating data suggest that epigenetic mechanisms may be regulated by various environmental factors, including dietary phytochemicals—substances of plant origin that possess biological activity. Moreover, there is an opinion that the selection of appropriate phytochemicals with known mechanisms of action may allow using them for the treatment of these disorders.

## 6. Phytochemicals

### 6.1. General Characteristics

Phytochemicals are structurally diverse biologically active compounds of plant origin that have important medicinal and nutritional properties. Phytochemicals are diverse in structure and affect various steps in the epigenetic mechanism. For example, polyphenols from green tea inhibit DNA methylation, while an isothiocyanate sulforaphane from broccoli sprouts, and diallyl disulfide from garlic are potent HDAC inhibitors [54,55,56]. In other studies, the epigenetic effect of polyphenol on gene expression is realized via miRNAs, which are considered global regulators controlling cell homeostasis and modulating inflammation [57]. Epidemiological studies indicate that the regular consumption of vegetables, fruits, and whole grains with high content of biologically active phytochemicals diminishes the risk of various diseases [58,59]. A wide range of plant origin substances providing health benefits includes flavonoids, polyphenols, anthocyanidins, carotenoids, phytoestrogens, terpenoids, phytosterols, fibers, and other compounds. Phytochemicals from medicinal plants often offer a better and safer alternative to synthetic medications. The anti-inflammatory, antioxidant, antiallergic, immunomodulatory and antitumoral activities of phytochemicals have been known for a long time; however, the molecular mechanisms of their action are often unclear and need further investigation. Accumulating data indicate that their biological activity is often associated with their role as regulators of gene expression. Phytochemicals may modulate specific genes’ transcriptional activity acting directly on specific TFs or exert their action through epigenetic mechanisms [60,61,62]. Here, we describe recent data dealing with the biological effects of phytochemicals on neurodegenerative processes in the three most common synucleinopathies: PD, DLB, and MSA.

### 6.2. Beneficial Effects of Phytochemicals: Numerous Substances, Various Mechanisms

Phytochemicals possess different mechanisms of action; some of them are targeted at specific routes affecting signaling pathways, while others control neurodegeneration via binding to TFs or through epigenetic mechanisms. A group of phytochemicals are able to upregulate the expression of cytoprotective genes [63]. There are phytochemicals whose activity is directed against common destructive pathways shared by synucleinopathies and other neurodegenerative diseases. For example, dietary saffron regulates apoptosis, inflammation and mitochondrial dysfunction by upregulating several genes (Cyr61, Gpx8, Ndufs4, and Nos1ap) with known neuroprotective actions [61]. Others, for example, natural polyphenols, are more specific and prevent the aggregation of α-synuclein and other prone-to-aggregate proteins [64,65], or regulate the expression of genes involved in PD, DLB and MSA pathogenesis [60,62].

The mechanisms of some of the phytochemicals are studied in detail, and specific molecular targets of their activity are identified, whereas for others multiple effects relevant to the neurodegenerative processes are detected. For example, polyphenol curcumin present in high quantity in turmeric herb *Curcuma longa* is an extremely strong antioxidant possessing in addition a strong epigenetic effect on gene expression [66].

Another group of polyphenols—catechins—display some neuroprotective activities by activation of protein kinase C (PKC) and via TFS that upregulate cell-survival genes [67]. Catechins and their metabolites activate multiple signaling pathways that exert anti-inflammatory and cell-survival activity, upregulating antioxidant pathways and modulating the expression of pro- and antiapoptotic proteins, and upregulating antioxidant defenses [67].

Methyl jasmonate [methyl 3oxo-2-(2-pentenyl) cyclopentaneacetate] (Figure 3) is an oxylipin—a derivative of oxygenated fatty acid synthesized from linolenic acid in chloroplast membranes from jasmine. It abrogates parkinsonianlike symptoms possessing multiple protective activities against neurodegeneration. Methyl jasmonate inhibits oxidative stress, has anti-inflammatory activity and down regulates α-synuclein expression [68].

Interestingly, jasmonate possesses beneficial effect not only in humans, but also in plants, helping them to adapt to external stressors by the regulation of stress responding genes. Jasmonates are vital phytohormones for plant survival and development. Jasmonic acid and its derivatives induce the synthesis of defensive chemical substances that protect plants against a wide range of stress [70,71]. Methyl jasmonate is a volatile substance that is also responsible for interplant communication. An area of further thrilling investigation is deciphering whether the protective activities exerted by jasmonate in plants and humans are realized via similar or different pathways and mechanisms.

The detailed molecular mechanism of jasmonate and other phytochemicals is not investigated in detail and needs further study. For example, their role in epigenetic mechanisms needs to be studied more thoroughly [72].

### 6.3. Phytochemicals and Epigenetic Regulation

A growing number of studies reveal that the effect of phytochemicals from medicinal plants are displayed via epigenetic mechanisms (Figure 2) by DNA methylation, histone modifications, microRNAs or LncRNAs [73].

For example, curcumin plays multiple roles as an epigenetic regulator, affecting many components at various steps of this process [66]. It regulates histone modifications via the modulation of histone acetyltransferases (HATs) and HDACs. Curcumin also inhibits the activity of DNMTs, regulates the level of specific microRNAs (miRNA), interacts with TFs and acts as a DNA binding agent [66]. In addition to curcumin, a group of polyphenols contains other phytochemicals with multiple beneficial properties. Polyphenols are widely present in cloves, berries, beans, nuts, etc. They regulate free radical production by several mechanisms, including the effect on the expression of the genes inhibiting oxidative stress and neutralizing free radicals [64,65].

Another biologically active phytochemical is melatonin (N-acetyl-5-methoxytryptamine). In addition to its antioxidant activity against reactive oxygen and nitrogen species, it also possesses anti-inflammatory properties and reduces the risk of PD [74,75,76]. Melatonin was known for a long time as an animal hormone for its role in circadian rhythms, immunological enhancement, regulation of mood, sleep, and sexual behavior. Its presence and physiological role in plants was described only in 1995 [77]. Melatonin has been detected in the leaves, roots, shoots, fruits, and seeds of many plant species.

#### Resveratrol

Resveratrol (Figure 4) is abundant in peanuts, red grapes, cacao beans, and many types of berries. It is also present in substantial amounts in red wine.

Resveratrol possesses multiple epigenetic effects, for instance, it inhibits the activity of enzymes controlling epigenetic processes, e.g., DNMTs, HDACs, or HATs [39]. Furthermore, resveratrol modifies the expression of several genes controlling overall health and longevity [78]. The epigenetic effect of resveratrol is also carried out through its effect on the expression of noncoding miRNAs, specifically, miR-132, miR-124, miR-134, miR-15a, and miR-146 [79]. Another important regulatory mechanism of resveratrol is upregulation of miR-let7A, which modulates the level of IL-6, TNF, IL-10, BDNF, and ASK1 [80].

Several studies identify that resveratrol and other dietary polyphenols exert their epigenetic regulation by modulating proinflammatory and anti-inflammatory miRNAs [57]. A more detailed investigation identified miR-663 as a resveratrol target that controls the TGFβ1 signaling pathway [81].

Below we present several examples of how phytochemicals might affect the three types of synucleinopathies (Section 6.4, Section 6.5 and Section 6.6).

### 6.4. Effect of Phytochemicals on PD Pathogenesis

Multiple studies related to PD using various model systems have shown that the beneficial properties of phytochemicals may be displayed through their effect on transcriptional machinery by changing gene expression levels. For example, epigallocatechin gallate (EGCG) significantly increases the protein expression of FOXO1, Sirt1, CAT, FABP1, GSTA2, ACSL1, and CPT2 and simultaneously reduces the levels of NF-κB, ACC1, and FAS protein. These gene expression changes decrease the level of inflammatory and oxidative stress and extend the lifespan in animal models [82].

In another study, phytochemicals extracted from purple yam (*Dioscorea alata L*) enhanced the expression of stress-protective genes such as Hsp-16.2, Hsp-6, Hsp-60, and Gst-4. These plant extracts acted via master TFs: HSF-1 and SKN-1/Nrf2 and regulated tic stress response, enhancing the expression of glyoxalase-1, heat shock, and antioxidant genes [83].

Menendez et al. described that secoiridoid-rich phenolic fractions from extra virgin olive oil (EVOO) activated resveratrol-like transcriptomic signatures [84]. Secoiridoids belong to a subclass of iridoids resulting from the cleavage of a bond in the cyclopentane ring. Secoiridoids from EVOO upregulated SIRT1 and inhibited several metallothionein (MT) gene isoforms.

Nutrigenomics is a quickly growing field studying the mechanisms of phytochemicals’ effects on the human genome [60]. Currently, an emerging trend combines the data from biochemistry, nutrition, genomics, transcriptomics, proteomics, metabolomics, and epigenomics to elucidate the existing mutual interactions between genes and nutrients at a molecular level.

### 6.5. Potentially Beneficial Effect of Phytochemicals for DLB Treatment

Currently, there is a growing interest in using phytochemicals as a therapeutic strategy for DLB treatment. Many potential plant-derived chemicals exert a neuroprotective effect via an epigenetic mechanism. However, little information is available about their mechanism of action. There are reasons to believe that this outcome may be similar to the effect on PD described in Section 6.2, since α-synuclein plays an essential role in both disorders.

A beneficial role of vitexin is described for DLB and other neurodegenerative diseases. Vitexin, found in several medicinal plants such as hawthorn, pearl millet, and *Ficus deltoidea*, regulates neuroinflammation and possesses a neuroprotective effect. Vitexin epigenetically regulates the proinflammatory cytokines and can be considered a potential candidate for the treatment of DLB and other neurodegenerative diseases [85].

Another example of phytochemicals with an epigenetic mechanism of action which has a beneficial effect for the treatment of DLB is resveratrol and other polyphenols [86].

A drawback of resveratrol is its low bioavailability, so it can be replaced by its glucoside polydatin, which is more abundant. Polydatin or piceid is a glucoside form of resveratrol with substitution of a hydroxyl group for glucose. The neuroprotective properties of polydatin against DLB, Parkinson’s disease dementia, and other dementia-related diseases have been described in several animal models [87]. Polydatin possesses a neuroprotective effect, attenuating neuronal oxidative damage and inflammation by activating MAPK and regulating the AKT/GSK3βNrf2/NF-κB signaling pathway [87].

Resveratrol regulates epigenetic changes that persist across generations [39]. The alterations of the epigenome cause changes in the transcriptional pattern and genomic instability associated with neurodegenerative diseases and other disorders.

### 6.6. Potentially Beneficial Effect of Phytochemicals for MSA Treatment 

The information about phytochemicals’ use for patients with MSA is rather limited. Sakakibara and coauthors [88] investigated herb extracts’ effects on patients with MSA and PD. They used Dai-Kenchu-To (DKT) consisting of a mixture of 50% ginger, 30% “Nin-jin” (gin-seng), and 20% “Sansho” (Japanese pepper, Zanthoxylum). DKT is a dietary herb extract used for relieving abdominal distension in Eastern countries. The DKT was well tolerated and improved gastrointestinal tract function both in PD and MSA patients. There were no noticeable differences between the MSA and PD groups in their response to DKT treatment.

In another study, strong antifibrillogenic and antioxidant activities of tannic acid, nordihydroguaiaretic acid, curcumin, rosmarinic acid, myricetin and other phytochemicals were demonstrated. The authors suggest using these substances to develop preventives and therapeutics for MSA and other neurodegenerative diseases [89].

The mechanism of action of these phytochemicals is not investigated in detail; however, since α-synuclein plays an essential role in MSA and PD, there are reasons to believe that these mechanisms are similar in these two diseases.

Gaps in our knowledge. There are many gaps in the current knowledge about the epigenetic activity of phytochemicals. First, there is a deficiency of human trials, which are essential because the human organism displays high complexity in responding to external stimuli. As a result, the findings from cell and animal models are often difficult to translate into human interventions. Another gap is the insufficient knowledge about the molecular mechanisms underlying epigenetics-related interactions. For example, α-synuclein in three forms of synucleinopathies is present in different conformational strains with dissimilar biochemical and biophysical properties [8]. It is unknown how different the effect of phytochemicals would be on these strains. Finally, phytochemicals research results are not thoroughly standardized, presenting obstacles for comparative studies to overcome.

## 7. Conclusions

The prevalence of synucleinopathies is rising, but efficient medications affecting these diseases have not been developed yet. Therefore, finding novel targets for their treatment and searching for new substances to improve conditions and cure these disorders are urgently needed. Emerging evidence suggests that plant-derived substances may be successfully used as an alternative to existing drugs. Phytochemicals target various elements in the pathogenesis of synucleinopathies, exerting a neuroprotective effect. Numerous studies revealed that healthy dietary practice with a high content of plant-derived substances boosts brain health, improves cognitive function, and suppresses neurodegenerative events [90,91,92,93]. Furthermore, the effect of numerous food-derived compounds from plants was tested in an attempt to improve the course of neurodegenerative diseases and to find an alternative to existing drugs. Problems with permeability often complicate the clinical research on the beneficial effects of phytochemicals through the brain–blood barrier (BBB), fast metabolism, and reduced stability inside the brain.

The perception of how the epigenetic mechanisms contribute to the pathogenesis of these diseases is a burning challenge, and the results of new investigations could have major importance for the development of future therapy. Epigenetics promises the explanation of many medical mysteries, including the etiology and pathophysiology of synucleinopathies. The study of the epigenetic mechanisms will serve as a useful tool for discovering specific phytochemicals to convert them to a valuable protective and even a therapeutic substitute for the treatment of synucleinopathies. Innovative studies should be performed as short-term projects analyzing various combinations of phytochemicals, in which biomarkers for disease outcome along with molecular markers (-omics) and genetic variability between individuals are incorporated.

## Figures and Tables

**Figure 1 biomolecules-11-00624-f001:**
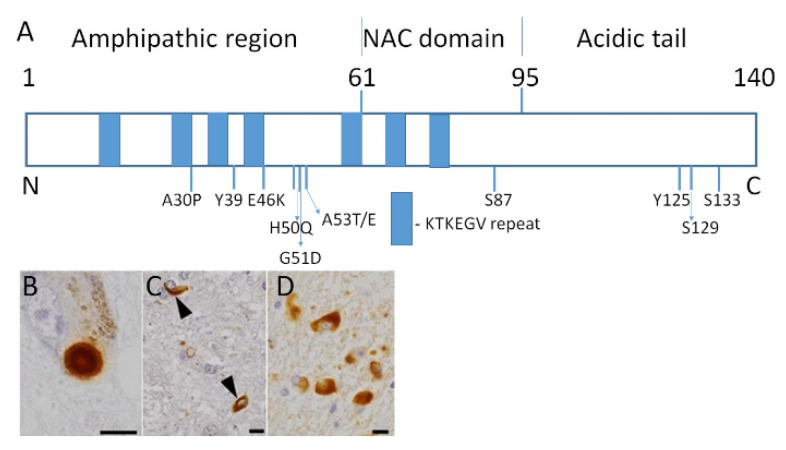
α-synuclein structure and depositions of misfolded protein in the brain. (**A**) Structural organization of the human α-synuclein. The N-terminal domain contains repeated lipid-binding sequences KTKEGV (blue rectangles). Mutations linked with familial cases of PD and sites of phosphorylation are shown. A30P, E46K, H50Q, G51D, A53T/E—point mutations in the α-synuclein gene. Y39, S87, Y125, S129, S133—sites of phosphorylation. The majority of α-synuclein in inclusions is phosphorylated. (**B)** Lewy body in substantia nigra of PD patients. (**C)** α-synuclein inclusions in DLB shown by arrowheads. (**D)** α-synuclein inclusions in MSA. Bars are 10 μm. Reproduced with permission from an article by Odagiri et al. [6] published in *Acta Neuropathologica* by Springer Nature, 2012, 124, 173–86. License number 5021401507585.

**Figure 2 biomolecules-11-00624-f002:**
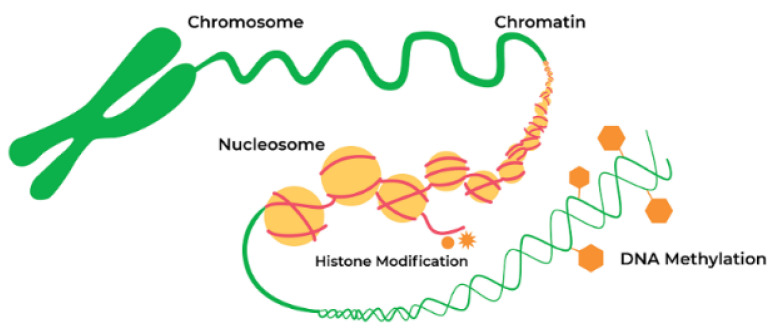
Epigenetic alterations are heritable and stable gene expression changes that occur through modifications in the chromosome rather than in the DNA sequence. Epigenetic mechanisms can regulate gene expression via chemical modifications of DNA bases and alterations to the chromosomal structure of DNA. Reproduced with permission from a website of “Zymo Research”, Available online: https://www.zymoresearch.com/pages/what-is-epigenetics (accessed on 20 April 2021).

**Figure 3 biomolecules-11-00624-f003:**
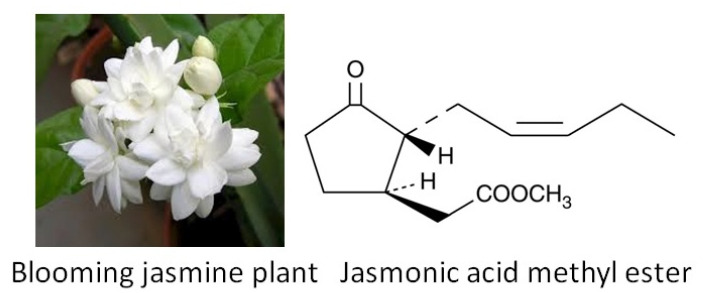
Jasmonic acid methyl ester isolated from jasmine flowers possesses neuroprotective effect in human brain and simultaneously helps plants to resist stress. Jasmonic acid signaling involves E3 ubiquitin ligase complexes, which tag substrates with ubiquitin. After tagging the proteins become susceptible for degradation by proteasomes. The subsequent step utilizes TFs to effect physiological changes. A key regulator in JA signaling is JAZ. Two TFs, MYC2 and ERF1, orchestrate subsequent changes in gene expression [69].

**Figure 4 biomolecules-11-00624-f004:**
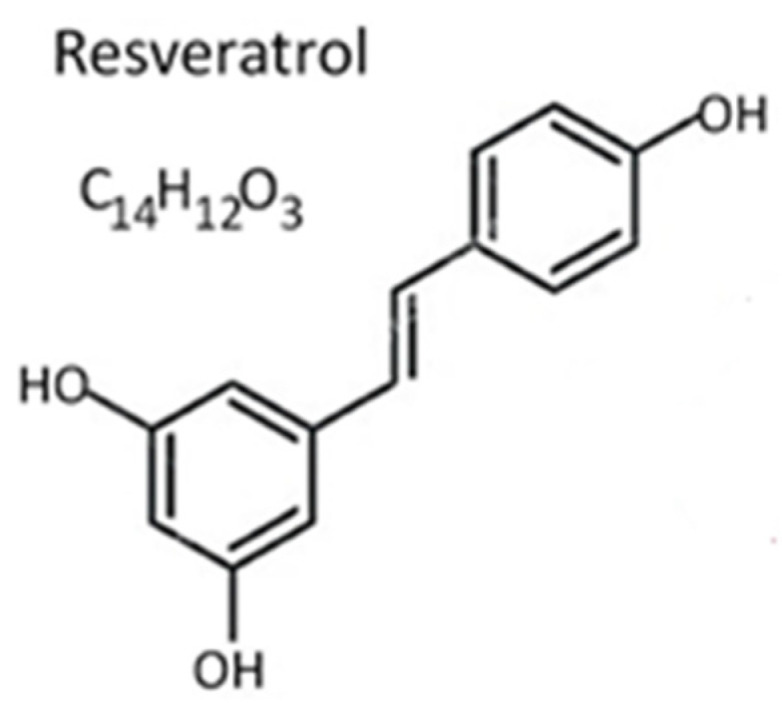
Resveratrol (3,4′,5-trihydroxy-trans-stilbene) is a natural polyphenol with high antioxidant and anti-inflammatory properties produced by some plants in response to injury.

## Data Availability

Not applicable for studies not involving humans.

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
