# Peer review of "Phytochemicals as Regulators of Genes Involved in Synucleinopathies"

_biomolecules, 2021, doi:10.3390/biom11050624_

Round 1

Reviewer 1 Report

The review is comprehensive and well structured on a very important topic that has not been reviewed in this way previously. I have a few minor suggestions:

  • formatting - please see lines 330 to 335 - there seems to be some text missing or it not formatted with the figure attached
  • Line 97 - can "Chapter" be changed to "Section" - otherwise it reads like an excerpt for a thesis
  • sections on phytochemicals reads differently (as if written by a different author) and read like a list of actions - can authors synthesise this knowledge into pathways in a pictorial form?
  • Line 468 - please remove "cure" from the sentence as the linked papers do not represent cured patient populations and are reviews; it is not appropriate to say that neurodegenerative diseases can be cured by specific food - there is evidence of action across different biological pathways but not cure 

If possible, I would also like to see some pictorial summary.

Author Response

We would like to thank reviewers for their comments and suggestions. We corrected the manuscript as recommended and we believe that these corrections allowed us to improve its. Below is a list of what we did:

Reviewer 1.

The review is comprehensive and well structured on a very important topic that has not been reviewed in this way previously. I have a few minor suggestions:

*formatting - please see lines 330 to 335 - there seems to be some text missing or it not formatted with the figure attached

-Thank you for letting us know. We corrected the legend to Figure 3 as follows :”Jasmonic acid methyl ester isolated from jasmine flowers possesses neuroprotective effect in human brain and simultaneously helps plants to resist stress. Jasmonic acid signaling involves E3 ubiquitin ligase complexes, which tag substrates with ubiquitin. After tagging the proteins become susceptible for degradation by proteasomes. The subsequent step utilizes TFs to effect physiological changes. A key regulator in JA signaling is JAZ, and two TFs, MYC2 and ERF1, orchestrate subsequent changes in gene expression [69]”.

* Line 97 - can "Chapter" be changed to "Section" - otherwise it reads like an excerpt for a thesis.

-Thank you, we corrected this and now it is named “Section 6”.

* sections on phytochemicals reads differently (as if written by a different author) and read like a list of actions - can authors synthesise this knowledge into pathways in a pictorial form?

_Yes, this is possible, since three contributing authors may have their own style. We combined our efforts and introduced some changes to improve the text and make it more pictorial.

*Line 468 - please remove "cure" from the sentence as the linked papers do not represent cured patient populations and are reviews; it is not appropriate to say that neurodegenerative diseases can be cured by specific food - there is evidence of action across different biological pathways but not cure

-We replaced the sentence “Furthermore, neurodegenerative diseases have been therapeutically cured by food-derived compounds from plants [91, 93]”  by the following “Furthermore, the effect of numerous food-derived compounds from plants was tested in an attempt to improve the course of neurodegenerative diseases and to find an alternative to existing drugs”.

*If possible, I would also like to see some pictorial summary

-Since we already have Abstract and Conclusion, we added several sentences to the Conclusion in response to this suggestion and in an attempt to make it more pictorial.  For example,

1) we added a sentence at the end of Conclusion “Innovative studies should be performed as short‐term projects analyzing  various combinations of phytochemicals, in which biomarkers for disease outcome along with molecular markers (-omics) and genetic variability between individuals are incorporated”

2) We also made some corrections, for example in the sentence beginning from “Furthermore, neurodegenerative diseases have been therapeutically cured….”

Reviewer 2 Report

The authors in this review article elaborated on the regulatory effects of phytochemicals by their ability to modulate the expression of genes involved in synucleinopathies.  This article would be certainly in the interest of the readers with progress in understanding a more detailed mechanism of the synucleinopathies disease regulation and novel therapeutic avenues. The following concerns can be addressed before the publication of the article. 1. Lines 91-93 statements should have references.  2.Line 115-117, the sentence is confusing. May need to rewrite. 3. Line 136-140, is this text needed in the context of the review? It may be part of a more detailed review related to epigenetics but seems unnecessary for this review or may need to be rewritten to connect with the following paragraph.  4. Figure 2, why is DNA covering nucleosome is having different colors? Any specific reason? 5. Line 153-154, if there any study previously reported to add as a reference for this statement then should be included.  6. Line 327, is this figure and its text complete? It seems something is missing or the sentence is confusing. “…in human brain and Scheme 3. ubiquitin ligase complexes, which tag substrates with..” 7. Line 366-367, seems this sentence is not correctly placed!

Author Response

Reviewer 2

We would like to thank reviewers for their comments and suggestions. We corrected the manuscript as recommended and we believe that these corrections allowed us to improve its. Below is a list of what we did:

The following concerns can be addressed before the publication of the article.

*1. Lines 91-93 statements should have references. 

Thank you, we added references after the sentence: “Nutrition, including consumption of phytochemicals, are essential environmental factors related to PD through epigenetic pathways” [16].

*2. Line 115-117, the sentence is confusing. May need to rewrite.

To exclude misunderstanding of this sentence we corrected it as follows :” There are no medications specific for the treatment of DLB approved by the US Food and Drug Administration (FDA). Therefore, drugs developed and tested for AD and PD treatment are sometimes used for DLB therapy [27,28].

*3. Line 136-140, is this text needed in the context of the review? It may be part of a more detailed review related to epigenetics but seems unnecessary for this review or may need to be rewritten to connect with the following paragraph. 

In response to this suggestion we truncated the majority of the sentence leaving only the most essential part:”Epigenetic modifications alter the gene function without changing the underlying DNA sequence [36, 37].

*4. Figure 2, why is DNA covering nucleosome is having different colors? Any specific reason?

This is done to show that DNA bound to nucleosomes changes its secondary structure  

*5. Line 153-154, if there any study previously reported to add as a reference for this statement then should be included. 

We added reference [39] after this sentence. “These epigenetic modifications can be induced by various factors, including phytochemicals present in the diet. Epigenetics changes in gene function are heritable and are not attributed to alterations of the DNA sequence [39].

*6. Line 327, is this figure and its text complete? It seems something is missing or the sentence is confusing.

The sentence may look confusing, because it was split during the conversion from Word to PDF. We fix it and now it looks like the end of legend to Figure 3:

Line 327 ”…which tag substrates with ubiquitin. After tagging the proteins become susceptible for degradation by proteasomes. The subsequent step utilizes TFs to effect physiological changes. A key regulator in JA signaling is JAZ, and two TFs, MYC2 and ERF1, orchestrate subsequent changes in gene expression [69].”

*7. Line 366-367, seems this sentence is not correctly placed!

Thank you, in response to this commentary we moved the end of this sentence further down to line 384 and now it looks like this: “Below we present several examples of how phytochemicals might affect the three types of synucleinopathies (sections 6-4, 6-5 and 6-6). “